# Prognosis Associated with Sub-Types of Hyperglycaemia in Pregnancy

**DOI:** 10.3390/jcm10173904

**Published:** 2021-08-30

**Authors:** Emmanuel Cosson, Sid Ahmed Bentounes, Charlotte Nachtergaele, Narimane Berkane, Sara Pinto, Meriem Sal, Hélène Bihan, Sopio Tatulashvili, Jean-Jacques Portal, Lionel Carbillon, Eric Vicaut

**Affiliations:** 1AP-HP, Avicenne Hospital, Department of Endocrinology-Diabetology-Nutrition, CRNH-IdF, CINFO, Paris 13 University, Sorbonne Paris Cité, 93000 Bobigny, France; narimane.berkane@aphp.fr (N.B.); meriem.sal@aphp.fr (M.S.); helene.bihan@aphp.fr (H.B.); sopio.tatulashvili@aphp.fr (S.T.); 2Nutritional Epidemiology Research Unit, UMR U557 INSERM/U11125 INRAE/CNAM, Paris 13 University, Sorbonne Paris Cité, 93000 Bobigny, France; 3AP-HP, Clinical Research Unit St-Louis-Lariboisière, Denis Diderot University, 75010 Paris, France; sidahmed_bentounes@yahoo.fr (S.A.B.); nachtergaele.charlotte@yahoo.fr (C.N.); jean-jacques.portal@aphp.fr (J.-J.P.); eric.vicaut@aphp.fr (E.V.); 4AP-HP, Jean Verdier Hospital, Unit of Endocrinology-Diabetology-Nutrition, CRNH-IdF, CINFO, Paris 13 University, Sorbonne Paris Cité, 93143 Bondy, France; sara.pinto@aphp.fr; 5AP-HP, Jean Verdier Hospital, Department of Obstetrics and Gynecology, Paris 13 University, Sorbonne Paris Cité, 93143 Bondy, France; lionel.carbillon@aphp.fr

**Keywords:** gestational diabetes mellitus, early screening, prognosis

## Abstract

We aimed to compare pregnancy outcomes in 4665 women according to the following types of hyperglycaemia in pregnancy sub-types: (i) normoglycaemia, (ii) gestational diabetes mellitus (GDM), (iii) diabetes in pregnancy (DIP), (iv) early-diagnosed (i.e., <22 weeks of gestation) GDM (eGDM), and (v) early-diagnosed DIP (eDIP). The prevalence of normoglycaemia, eGDM, eDIP, GDM, and DIP was 76.4%, 10.8%, 0.6%, 11.7%, and 0.6%, respectively. With regard to pregnancy outcomes, gestational weight gain (11.5 ± 5.5, 9.0 ± 5.4, 8.3 ± 4.7, 10.4 ± 5.3, and 10.1 ± 5.0 kg, *p* < 0.0001) and insulin requirement (none, 46.0%, 88.5%, 25.5%, and 51.7%; *p* < 0.001) differed according to the glycaemic sub-types. eGDM and eDIP were associated with higher rates of infant malformation. After adjustment for confounders, with normoglycaemia as the reference, only GDM was associated with large-for-gestational-age infant (odds ratio 1.34 (95% interval confidence 1.01–1.78) and only DIP was associated with hypertensive disorders (OR 3.48 (1.26–9.57)). To conclude, early-diagnosed hyperglycaemia was associated with an increased risk of malformation, suggesting that it was sometimes present at conception. Women with GDM, but not those with eGDM, had an increased risk of having a large-for-gestational-age infant, possibly because those with eGDM were treated early and therefore had less gestational weight gain. Women with DIP might benefit from specific surveillance for hypertensive disorders.

## 1. Introduction

Hyperglycaemia in pregnancy (HIP) is a common condition associated with poor maternal and neonatal outcomes [1,2]. Its diagnosis excludes any known diabetes before pregnancy. Both gestational diabetes mellitus (GDM) and diabetes in pregnancy (DIP) contribute to HIP [3,4]. DIP was introduced as a medical category by the International Association of Diabetes Pregnancy Study Group (IADPSG) after experts recommended that women who have high glycaemic levels when screened during pregnancy be considered as having unknown type 2 diabetes prior to pregnancy [3]. DIP is diagnosed using the same glucose threshold (fasting plasma glucose (FPG) ≥ 7 mmol/L) and/or HbA1c value (≥6.5% (48 mmol/mol)) used to diagnose diabetes in non-pregnant individuals [2,3,4].

Hyperglycaemic states usually appear late in the second trimester of pregnancy, when insulin resistance increases [5], and screening is typically performed at this time. However, as DIP is considered as unknown type 2 diabetes before pregnancy, the IADPSG and the World Health Organisation (WHO) initially recommended screening for it in early pregnancy, as earlier care could improve its prognosis [2,3,4]. It could also lead to the identification of women with early intermediate hyperglycaemia. This initial recommendation [3,4] was, however, reconsidered by both organisations for two main reasons [6]: first, although treating GDM after 24 weeks of gestation (WG) reduces adverse events during pregnancy [7,8], the same has not been demonstrated for early-diagnosed GDM (eGDM) [6,9]. Second, at least half of untreated women with early fasting hyperglycaemia do not develop GDM later in pregnancy [10,11,12,13,14]. Currently, no data exist in the literature on the persistence of DIP throughout pregnancy.

In 2010, in line with the then current recommendations from the IADPSG, French guidelines recommended screening for HIP with a fasting plasma glucose (FPG) measurement during the first trimester [2]. Women with an FPG of 5.1 to 6.9 mmol/L and 7 mmol/L or more are diagnosed with eGDM and “early-diagnosed” DIP (eDIP), respectively. They receive appropriate care and are not screened again during pregnancy. For those with an FPG lower than 5.1 mmol/L, an oral glucose tolerance test (OGTT) is performed between the second semester trimester and early in the third trimester (Figure 1) [2].

The primary aim of the present study was to compare pregnancy outcomes according to five different HIP sub-types in a cohort of women for whom this strategy was used [10,15,16]. The HIP sub-types were as follows: normoglycaemia, eGDM, eDIP, late-diagnosed (i.e., >22 WG) GDM (indicated as GDM throughout this manuscript), and late-diagnosed DIP (DIP throughout the manuscript). The secondary aim was to evaluate whether the association between gestational weight gain and pregnancy outcomes differed according to each HIP sub-type.

## 2. Materials and Methods

### 2.1. Participants

This observational study was conducted in Jean Verdier University hospital located in Bondy, a suburb of Paris, France. It was based on electronic medical records of women who delivered babies between 2012 and 2016. Obstetrical data were prospectively, routinely, and systematically collected at birth in our hospital by the midwife assisting with the delivery. In addition, we collected data on hyperglycaemia screening of these mothers [10,15,16]. Inclusion criteria were being aged between 18 to 50 years, singleton pregnancy, and no personal history of diabetes prior to pregnancy or prior bariatric surgery. We selected women who had an FPG measurement taken before 22 WG, and those who subsequently had an OGTT between 22 and 30 WG, if their FPG was lower than 5.1 mmol/L (flowchart in Appendix A).

### 2.2. Definitions of Glycaemic Status during Pregnancy and Management of Hyperglycaemia

Our hospital’s policy was to screen all pregnant women both at the beginning of pregnancy and after 22 WG if prior screening was normal or not performed. HIP was defined according to IADPSG/WHO criteria [3,4] which were endorsed in France in 2010 [2] (Figure 1).

Women with HIP were referred to our multidisciplinary team comprising a diabetologist, an obstetrician, a midwife, a dietician, and a nurse educator within 12 weeks after diagnosis. In line with French recommendations [2], they received individualized dietary advice and instructions on how to perform self-monitoring of blood glucose levels six times a day. They were also seen by the diabetologist every 2–4 weeks and received insulin therapy when their pre-prandial and/or 2 h post-prandial glucose levels were greater than 5.0 and/or 6.7 mmol/L, respectively, according to the French guidelines [2]. Obstetrical care also followed French recommendations [2]. Women with normoglycaemia did not receive any specific care.

### 2.3. Pregnancy Outcomes

We considered two sets of outcomes [10,15,16,17,18], one categorized as “maternal” and the other “neonatal” (as defined by the INSPIRED research group). Maternal outcomes included HIP diagnosis, hypertensive disorders, pharmacological therapy for HIP, and mode of child delivery. Neonatal outcomes included birth weight, large-for-gestational-age (LGA) infant, small-for-gestational-age infant, gestational age at birth, preterm birth, neonatal hypoglycaemia, neonatal death/stillbirth, and malformation [19].

Gestational hypertension was defined as the onset of hypertension (blood pressure ≥140 mmHg systolic or ≥90 mmHg diastolic) at or after 20 WG, in the absence of proteinuria and without biochemical or haematological abnormalities. When earlier blood pressure values were unknown, they were considered to be normal. Preeclampsia was defined as having a blood pressure ≥ 140/90 mmHg for two measurements four hours apart, and proteinuria of at least 300 mg/24 h or a 3+ level with dipstick testing in a random urine sample, and/or evidence of maternal acute kidney injury, liver dysfunction, neurological features, haemolysis or thrombocytopenia, and/or foetal growth restriction [20]. Hypertensive disorders were defined as having gestational hypertension and/or preeclampsia. We also considered selective and emergency (i.e., before or during delivery) Caesarean sections.

LGA and small-for-gestational-age were defined as a birth weight greater than the 90th percentile and lower than the 10th percentile for a standard French population, respectively [10,14,21,22]. Preterm birth was defined as occurring before 37 WG and neonatal hypoglycaemia as having at least one blood glucose measurement under 2 mmol/L during the first two days of life. Finally, we also considered neonatal deaths (in the first 24 h of life) and stillbirths.

### 2.4. Statistics

Continuous variables were expressed as means ± standard deviation. Categorical variables were expressed as frequencies (percentages). Gaussian distribution of the variables was graphically assessed and tested using the Kolmogrov–Smirnov test.

We compared the characteristics of the included and non-included women using Student’s *t*-test for continuous variables, and Pearson’s chi-squared (X2) for categorical variables.

We also compared the characteristics and events of the HIP sub-types (eGDM, eDIP, GDM, DIP, and normoglycaemia considered as the reference) using the ANOVA for continuous variables, and chi-squared (X2) or Fisher’s exact test for categorical variables. When an overall difference was found—with an a priori determined risk α = 5%—two-by-two comparisons were performed (i.e., normoglycaemia versus each HIP group, GDM versus DIP, and eGDM versus GDM). For each comparison, the α value was adjusted for multiplicity using Bonferroni correction.

Multivariable logistic regression analysis was used to estimate the probability of a binary event, hypertensive disorders, and LGA infant, and for the different sub-types after adjustment for the variables a priori known to potentially affect the risk of these events, i.e., age, gestational weight gain, insulin therapy, smoking during pregnancy, occupational status, and ethnicity.

In addition, we specifically explored the effect of gestational weight gain on hypertensive disorders during pregnancy and LGA infants in the HIP population using univariate and multivariate logistic regression. The interaction between HIP sub-type and weight gain was introduced into the model to test the hypothesis that the effect of gestational weight gain could differ according to HIP sub-type. All tests were conducted using SAS version 9.4. (from SAS institute) or R.

## 3. Results

### 3.1. Population Characteristics

Of the 11,234 pregnant women recorded in our hospital during the study period, 4665 were included in this study (Flowchart, Appendix A). The study population had an overall higher prevalence of risk factors for HIP than the 6569 women not included (the latter not having fully followed the recommended screening strategy; Appendix A). In terms of HIP sub-type, 3563 women had normoglycaemia (76.4%), 502 eGDM (10.8%), 26 eDIP (0.6%), 545 GDM (11.7%), and 29 DIP (0.6%).

Their baseline characteristics are shown in Table 1. Age and BMI differed according to HIP status. Specifically, women with eGDM, eDIP, and GDM were older than those without HIP, and approximately one third of the women with HIP had obesity. Table 1 also shows family history of diabetes, parity, employment, and history of HIP, LGA infant, hypertensive disorders, and foetal death during previous pregnancy according to HIP sub-type. Finally, ethnicity differed according to HIP status. For example, a large proportion of women from North Africa had HIP, while a large proportion of women from India, Pakistan, and Sri Lanka had DIP.

### 3.2. Maternal Outcomes According to HIP Status

Gestational weight gain and insulin therapy differed according to HIP status. The former was statistically lower in women with eGDM, eDIP, and GDM than in women with no HIP; it was lower in women with eGDM than in those with GDM. Insulin therapy was more frequent in women with eGDM (46.0%) and with DIP (88.5%) than in those with GDM (25.5%); 51.7% of women with eDIP were on insulin.

Caesarean section was more frequent in women with HIP; women with eGDM had a statistically higher rate (27.7%) than women without HIP (21.1%).

Hypertensive disorders occurred more frequently in women with DIP (20.7%) and eGDM (7.0%) than in women without HIP (4.2%); they were more frequent in women with DIP than those with GDM (3.5%). In the multivariable analysis, women with DIP were largely more likely to have hypertensive disorders during pregnancy than women with no HIP (Table 2: OR 3.48 (95% confidence interval 1.3–9.6)).

### 3.3. Neonatal Outcomes According to HIP Status

LGA infants were more frequent in women with GDM (13.9%) than in women without HIP (9.1%). This increased risk was confirmed in the multivariable analysis (Table 3: OR 1.34 (1.01–1.78)).

Preterm delivery was more frequent in women with eDIP (15.4%) than in women without HIP (4.9%); it was common in women with DIP (17.2%). Neonatal hypoglycaemia occurred more frequently in those with GDM (1.5%) and DIP (10.3%) than in those with no HIP (0.3%). No case of neonatal hypoglycaemia occurred in women with eDIP. Finally, the rate of infant malformation was higher in women with eDIP (7.7%) and eGDM (2.2%) than in women with no HIP (1.0%).

### 3.4. Hypertensive Disorders during Pregnancy and LGA Infant According to Gestational Weight Gain in Women with HIP

In this part of the study, we focused on women with HIP as they were provided care for their condition. We examined the role of gestational weight gain in two pregnancy outcomes: hypertensive disorders and LGA infants.

Figure 2 shows the proportion of women with HIP who had hypertensive disorders during pregnancy (panel A) and LGA infant (panel B) according to gestational weight gain. In these women, gestational weight gain was positively associated with LGA infant (panel B). In the multivariable analyses, the effect of gestational weight gain was not different for all HIP sub-types (*p* value for interaction > 0.05).

## 4. Discussion

### 4.1. Main Results

In our observational study, we selected pregnant women who had all been screened for HIP before 22 WG, and then after 22 WG when the first screening was normal. Using this complete screening strategy, we found that approximately one quarter of the women were diagnosed with HIP. In the latter, receiving care was associated with lower gestational weight gain than in women without HIP. In 25 to 88% (depending on sub-type) of those with HIP, glucose was controlled using insulin therapy. Despite this, classic HIP-related adverse outcomes were more frequent in those with HIP than their “no HIP” counterparts [1,18].

In the present study, we focused on the prognosis of HIP sub-types and found the following: (i) GDM was independently associated with LGA infant, (ii) DIP was an independent predictor of hypertensive disorders during pregnancy, (iii) 10% of neonates born to women with DIP experienced neonatal hypoglycaemia, and (iv) women with early-diagnosed HIP were associated with a higher risk of malformation than women with no HIP. Finally, the association between gestational weight gain and LGA infants in women with HIP did not differ according to HIP sub-type.

### 4.2. Prevalence of HIP Sub-Types

In line with the findings from another study conducted at our hospital [15], we found a high prevalence of HIP when screening was performed in early pregnancy and was then repeated during the second part of the pregnancy, when the first screening tested normal. Studies in various countries found similar percentages to ours of pregnant women with FPG values > 5.1 mmo/L in early pregnancy (10% in Israel [23], China [12], and Italy [11]). The percentage of women with DIP was relatively low in our series, unlike the findings in Pakistan [24] and Brazil [25], suggesting that screening for hyperglycaemia before pregnancy is probably performed more routinely in France.

### 4.3. Is Diabetes in Pregnancy Different between Early and Late Pregnancy?

As eDIP and DIP are considered to represent unknown type 2 diabetes, and therefore are supposed to remain present through pregnancy, we expected to have few cases of DIP as they would have been diagnosed as eDIP, thanks to our early screening strategy. Surprisingly, we found a similar prevalence of eDIP and DIP. Screening during early pregnancy was performed by FPG measurement alone, whereas later screening used OGTT, which includes 1-h and 2-h glucose values in addition to the FPG measurement. OGTT is the reference measure for HIP. FPG sensitivity to diagnose hyperglycaemia is approximately 50% both outside [26] and during [18] pregnancy. Accordingly, this suggests that when DIP is suspected, screening with FPG is insufficient and must be substituted by OGTT and/or HbA1c measurement. Two studies elsewhere showed that type 2 diabetes is confirmed by postpartum OGTT in 21.1% [27] to 47.5% [28] of women diagnosed with diabetes in pregnancy, with additional prediabetes identified in 32.5% to [28] 37.6% of cases [27]. Therefore, one can hypothesize that eDIP would correspond with pre-diabetic states prior to pregnancy, and that late-diagnosed DIP (with an initial normal FPG level) would correspond to some other mechanism, which would require further investigation. The differing prognosis associated with eDIP and with DIP that we found here is concordant with this hypothesis.

Two of the 26 women (prevalence 7.7%) with eDIP in our study had a child with a malformation. None of the 29 women with late-diagnosed DIP had one. Corrado et al. also found a high prevalence (9.4%) of infant malformation in women with diabetes in pregnancy (early and late diagnosis combined) [11]. Just as in our study, women with early-diagnosed HIP had a higher rate of infant with malformation than women without HIP. This reflects the known deleterious role poor glycaemic control before and in early pregnancy has on organogenesis. In our study, the rate of malformation was probably underestimated as we only included women who delivered babies after 22 WG (we could not consider possible miscarriages due to organogenesis issues).

Approximately 20% of the women with DIP in our study had hypertensive disorders versus 10% of those with eDIP. Just as is the case for LGA infants and neonatal hypoglycaemia, in persons with GDM, hypertensive disorders can be reduced by glycaemic control [6,7]. Looking at the large difference in the rates of insulin therapy between the women in our study with DIP (circa 50%) and those with eDIP (circa 80%), one might suppose that care providers hesitated before beginning insulin therapy when DIP was discovered late in pregnancy, leading to a poor glycaemic control in this sub-group, and therefore hypertensive disorders. However, this seems unlikely because the rate of LGA infant did not differ between HIP sub-types. The higher risk of hypertensive disorders could also illustrate the role of confounding factors, such as obesity; however, the association remained after multivariable analysis. This suggests that women with DIP would benefit from tailored surveillance of hypertensive disorders during pregnancy.

### 4.4. Are There Arguments for Screening for HIP in Early Pregnancy?

The high rate of infant malformations in women with eDIP illustrates the importance of screening women at high risk of hyperglycaemia before they become pregnant [26]. However, as pre-pregnancy screening is not routinely performed, screening at the start of pregnancy is recommended, even if malformations due to poor glycaemic control during organogenesis (the first 12 weeks of pregnancy) cannot always be avoided. The decision to screen early for HIP is usually based on the presence of risk factors for type 2 diabetes. In line with other studies, we found that increasing age [24,27,29], obesity [27,28,29,30], a family history of diabetes [11,29], a personal history of HIP [29], and ethnicity [29] were all associated with early-diagnosed HIP.

High FPG values in early pregnancy, without immediate consequent specific care, were previously associated with a greater risk of LGA infants [23,31]. One might suppose that immediate care for eGDM and eDIP would improve prognosis, but this had not been proven at that time [6,9,15,32,33]. Our results show that women with early-diagnosed HIP had a lower gestational weight gain than women with HIP diagnosed after 22 WG. Furthermore, they often needed insulin therapy to control glucose values, as previously reported [9]. In the multivariable analysis, we found that GDM was the only HIP sub-type to be independently associated with a higher risk of LGA infants (reference was normoglycaemia). The same rate of LGA infant in women with eGDM as in those without HIP advocates for the early identification of women with HIP and immediate care. In this study, gestational weight gain was an independent determinant of LGA infant, irrespective of HIP sub-types. However, observational studies have shown conflicting results [15,32,33,34,35,36], and one randomized study did not show any benefit of such early screening [37], or indeed of providing immediate care to women with early-diagnosed HIP [38].

### 4.5. Study Strengths and Limitations

The strengths of our study include (i) a large study sample, (ii) a multi-ethnic cohort (which suggests that our results are transferrable to different populations), and (iii) a pragmatic guidance-based approach using IADPSG/WHO criteria to define HIP. With regard to the latter, the prospectively collected standardized data provided a robust investigational data set, enabling us to adjust our results for a large set of confounders. Furthermore, we included women who had been screened both in early and middle pregnancy if the first screening was negative. This approach differed from that in observational studies that compared women with eGDM and GDM [9]. Very often, the latter (i.e., women with GDM) had only been screened in the second half of their pregnancy, which is problematic for two reasons: first, some of them would have already screened positive for eGDM had they been screened earlier [10,11,12,13,14] and therefore they would have been classified as eGDM using our methodology, and second, early screening in these studies is usually performed only in women with risk factors for type 2 diabetes/HIP. As risk factors, and especially obesity [39], are intrinsically associated with a poor prognosis (irrespective of HIP or not) [16], these studies suffered from inclusion bias, with a higher risk at inclusion in women with eGDM than in those with GDM. We had limited ability to evaluate—and even less to compare—the prognosis associated with eDIP and DIP. However, this study is the first to describe DIP diagnosed after 22 WG, subsequent to normoglycaemia screening (i.e., an FPG lower than 5.1 mmol/L) in early pregnancy.

## 5. Conclusions

We showed that pregnant women’s characteristics and pregnancy outcomes differed according to their HIP sub-types, the latter being defined according to time of diagnosis (before or after 22 WG) and glycaemic values during screening (which determined GDM or DIP). Women with early-diagnosed GDM and DIP, and especially those with eDIP, were more likely to have an infant with malformation, suggesting that screening for hyperglycaemia prior to pregnancy and subsequent care could help reduce this risk. Furthermore, women with late-diagnosed GDM, but not those with eGDM, had an increased risk of LGA infant after adjustment for confounders. Gestational weight gain was an independent determinant of LGA infant. Women with eGDM in our study had less gestational weight gain. These results suggest that care for early-diagnosed HIP may help limit the rate of LGA infants. Finally, women with DIP had an independent greater risk of hypertensive disorders (20% of this group). This finding may encourage care providers to improve surveillance of hypertensive disorders during pregnancy in persons diagnosed with DIP. Tailor-made therapeutic studies are essential to evaluate all these hypotheses in depth.

## Figures and Tables

**Figure 1 jcm-10-03904-f001:**
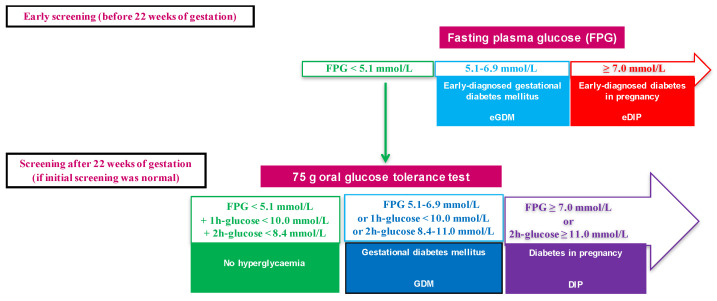
Screening for hyperglycaemia in pregnancy with diagnostic criteria.

**Figure 2 jcm-10-03904-f002:**
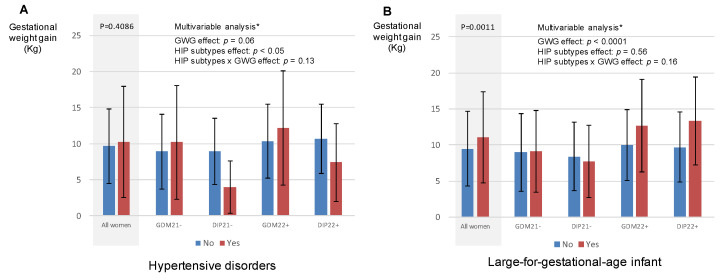
Pregnancy outcomes according to gestational weight gain and need for insulin in women with hyperglycaemia in pregnancy. (**A**) Hypertensive disorders according to gestational weight gain; (**B**) large-for-gestational infant according to gestational weight gain. WG—weeks of gestation; DIP—late-diagnosed (i.e., ≥22 weeks of gestation) diabetes in pregnancy; eDIP—early-diagnosed (i.e., <22 weeks of gestation) diabetes in pregnancy; eGD—early-diagnosed gestational diabetes mellitus; GD—late-diagnosed gestational diabetes mellitus; GWG—gestational weight gain; HIP—hyperglycaemia in pregnancy. * accounting for age, body mass index before pregnancy, parity, HIP sub-type, ethnicity, smoking during pregnancy, employment, GWG, need for insulin during pregnancy, and interaction between HIP sub-type and GWG.

**Table 1 jcm-10-03904-t001:** Characteristics and pregnancy outcomes according to glycaemic status.

	Available Data	Total	No HIP	eGDM	eDIP	GDM	DIP	Global Comparison *p*-Value
		*n* = 4665	*n* = 3563	*n* = 502	*n* = 26	*n* = 545	*n* = 29	
Screening for HIP before 22 WG								
Fasting plasma glucose (mmol/L)	*n* = 4238	4.6 (0.6)	4.5 (0.3)	5.4 (0.4) ^ab^	11.6 (3.4) ^a^	4.6 (0.3) ^a^	4.6 (0.3)	<0.0001
Gestational age at HIP screening (WG)	*n* = 4258	12.2 (4.2)	12.2 (4.2)	11.7 (4.5)	11.6 (0.7)	12.6 (4.1)	13.7 (4.5)	0.007
Screening with OGTT at 22 WG or later								
Fasting plasma glucose (mmol/L)	*n* = 3760	4.4 (0.5)	4.3 (0.4)	-	-	4.9 (0.6) ^a^	5.3 (1.0) ^ac^	<0.0001
1-h plasma glucose (mmol/L)	*n* = 3728	7.0 (1.8)	6.6 (1.5)	-	-	9.3 (1.7) ^a^	11.0 (2.4) ^ac^	<0.0001
2-h plasma glucose (mmol/L)	*n* = 3734	6.1 (1.5)	5.8 (1.1)	-	-	8.0 (1.6) ^a^	11.6 (1.9) ^ac^	<0.0001
Gestational age at OGTT (WG)	*n* = 4033	27.3 (3.0)	27.2 (3.0)	-	-	27.7 (3.3) ^a^	27.6 (3.6)	<0.0001
Metabolic characteristics								
Age (years)	*n* = 4665	30.7 (5.5)	30.2 (5.4)	32.6 (5.3) ^a^	33.3 (5.4) ^a^	32.1 (5.5) ^a^	32.6 (5.2)	<0.0001
Pre-pregnancy body mass index (kg/m^2^)	*n* = 4515	25.2 (5.1)	24.7 (4.8)	28.0 (5.9) ^ab^	27.7 (6.6) ^a^	26.2 (5.4) ^a^	27.7 (3.8) ^ac^	<0.0001
Pre-pregnancy obesity	*n* = 4515	833 (18.5)	531 (15.4)	170 (34.9) ^ab^	8 (32.0)	115 (21.9) ^a^	9 (33.3)	<0.0001
Pre-pregnancy hypertension	*n* = 4665	39 (0.9)	20 (0.6)	9 (1.8)	1 (3.9)	9 (1.7)	0 (0)	0.0027
Family history of diabetes	*n* = 4665	1291 (27.7)	892 (25.0)	204 (40.6) ^ab^	12 (46.2)	175 (32.1) ^a^	8 (27.6)	<0.0001
Employment at beginning of pregnancy	*n* = 4656	1887 (40.5)	1483 (41.7)	167 (33.3) ^a^	9 (34.6)	217 (39.9)	11 (37.9)	0.0097
Parity	*n* = 4665	2.1 (1.2)	2.1 (1.2)	2.3 (1.3) ^a^	2.5 (1.7)	2.2 (1.3)	1.8 (0.8)	0.0001
Previous pregnancy								
History of HIP	*n* = 4665							<0.0001 *
First child		1814 (38.9)	1452 (40.8)	147 (29.3)	7 (26.9)	195 (35.8)	13 (44.8)	
No		2561 (54.9)	2005 (56.3)	260 (51.8)	10 (38.5)	273 (50.1)	13 (44.8)	
Yes		290 (6.2)	106 (3.00)	95 (18.9)	9 (34.6)	77 (14.1)	3 (10.3)	
History of large for gestational age infant	*n* = 4665							<0.0001 *
First child		1814 (38.9)	1452 (40.8)	147 (29.3)	7 (26.9)	195 (35.8)	13 (44.8)	
No		2688 (57.6)	2012 (56.5)	326 (64.9)	15 (57.7)	322 (59.1)	13 (44.8)	
Yes		163 (3.5)	99 (2.8)	29 (5.8) ^a^	4 (15.4) ^a^	28 (5.1) ^a^	3 (10.3)	
History of hypertensive disorders	*n* = 4665							0.0104 *
First pregnancy		1265 (27.1)	1018 (28.6)	96 (19.1)	6 (23.1)	139 (25.5)	6 (20.7)	
No		3281 (70.3)	2462 (69.1)	383 (76.3)	18 (69.2)	395 (72.5)	23 (79.3)	
Yes		119 (2.6)	83 (2.3)	23 (4.6) ^a^	2 (7.7)	11 (2.0)	0 (0)	
History of foetal death	*n* = 4665							0.0418 *
First pregnancy		1265 (27.1)	1018 (28.6)	96 (19.1)	6 (23.1)	139 (25.5)	6 (20.7)	
No		3307 (70.9)	2485 (69.7)	391 (77.9)	19 (73.1)	391 (71.7)	21 (72.4)	
Yes		93 (2.0)	60 (1.7)	15 (3.0)	1 (3.9)	15 (2.8)	2 (6.9)	
Smoking during pregnancy	*n* = 4665	284 (6.1)	245 (6.9)	22 (4.4)	1 (3.9)	16 (2.9) ^a^	0 (0)	0.0012
Ethnicity	*n* = 4656							<0.0001
European		1269 (27.3)	1042 (29.3)	94 (18.8)	3 (11.5)	121 (22.2)	9 (31.0)	
African		881 (18.9)	709 (19.9)	80 (16.0)	5 (19.2)	83 (15.3)	4 (13.8)	
North African		1378 (29.6)	970 (27.3)	192 (38.3)	10 (38.5)	196 (36.0)	10 (34.5)	
Asian		96 (2.1)	70 (2.00)	10 (2.00)	1 (3.9)	14 (2.6)	1 (3.5)	
Caribbean		270 (5.8)	222 (6.2)	23 (4.6)	1 (3.9)	24 (4.4)	0 (0)	
Indian-Pakistan-Sri Lankan		519 (11.2)	346 (9.7)	82 (16.4)	6 (23.1)	80 (14.7)	5 (17.2)	
Other		243 (5.2)	197 (5.5)	20 (4.0)	0 (0)	26 (4.8)	0 (0)	
Maternal outcomes								
Gestational weight gain (kg)	*n* = 4316	11.1 (5.5)	11.5 (5.5)	9.0 (5.4) ^ab^	8.3 (4.7) ^a^	10.4 (5.3) ^a^	10.1 (5.0)	<0.0001
Insulin therapy during pregnancy	*n* = 1100	408 (37.1)	-	231 (46.0) ^b^	23 (88.5)	139 (25.5)	15 (51.7) ^c^	<0.0001
Caesarean section	*n* = 4665	1038 (22.3)	750 (21.1)	139 (27.7) ^a^	7 (26.9)	131 (24.0)	11 (37.9)	0.0019
Gestational hypertension	*n* = 4665	126 (2.7)	87 (2.4)	26 (5.2) ^ab^	2 (7.7)	8 (1.5)	3 (10.3) ^ac^	<0.0001
Preeclampsia	*n* = 4665	90 (1.9)	66 (1.9)	9 (1.8)	1 (3.9)	11 (2.0)	3 (10.3) ^a^	0.0211
Hypertensive disorders during pregnancy	*n* = 4665	214 (4.6)	151 (4.2)	35 (7.0) ^a^	3 (11.5)	19 (3.5)	6 (20.7) ^ac^	<0.0001
Neonatal outcomes								
Birthweight (g)	*n* = 4665	3294 (515)	3285 (508)	3335 (522)	3271 (573)	3321 (541)	3271 (643)	0.1984
Large-for-gestational-age infant	*n* = 4665	468 (10.0)	325 (9.1)	60 (12.0)	4 (15.4)	76 (13.9) ^a^	3 (10.3)	0.0039
Small-for-gestational-age infant	*n* = 4665	448 (9.6)	347 (9.7)	48 (9.6)	1 (3.9)	48 (8.8)	4 (13.8)	0.7259
Gestational age at birth (weeks of gestation)	*n* = 4665	39.7 (1.6)	39.7 (1.6)	39.5 (1.6)	38.7 (1.9) ^a^	39.4 (1.6) ^a^	39.0 (2.1)	<0.0001
Preterm delivery (<37 weeks of gestation)	*n* = 4665	259 (5.6)	173 (4.9)	38 (7.6)	4 (15.4) ^a^	39 (7.2)	5 (17.2)	0.0002
Neonatal hypoglycaemia	*n* = 4665	29 (0.6)	12 (0.3)	6 (1.2) ^a^	0 (0)	8 (1.5) ^a^	3 (10.3) ^a^	<0.0001
Neonatal death and stillbirth	*n* = 4665	18 (0.4)	14 (0.4)	2 (0.4)	0 (0)	2 (0.4)	0 (0)	0.0669
Malformation	*n* = 4665	51 (1.1)	32 (1.0)	11 (2.2) ^a^	2 (7.7) ^a^	6 (1.1)	0 (0)	0.0014

Data are *n* (%) or mean (standard deviation). DIP—late-diagnosed (i.e., ≥22 weeks of gestation) diabetes in pregnancy; eDIP—early-diagnosed (i.e., <22 weeks of gestation) diabetes in pregnancy; eGDM—early-diagnosed gestational diabetes mellitus; GDM—late-diagnosed gestational diabetes mellitus; HIP—hyperglycaemia in pregnancy. ^a^ Significant difference after Bonferroni correction for the comparison of each group of HIP with no-HIP. ^b^ Significant difference after Bonferroni correction for the comparison between eGDM and GDM. ^c^ Significant difference after Bonferroni correction for the comparison between DIP and GDM. * “Yes” versus “No” or “First child”.

**Table 2 jcm-10-03904-t002:** Determinants of hypertensive disorders in multivariable analysis.

	Odds Ratio	95% Confidence Interval	*p*-Value
Glycaemic status			
No HIP	REF		
eGDM	1.29	0.86–1.95	0.2164
eDIP	2.16	0.61–7.68	0.2352
GDM	0.67	0.40–1.11	0.1205
DIP	3.48	1.26–9.57	0.0159
Pre-pregnancy body mass index (kg/m^2^)	1.09	1.06–1.11	<0.0001
Age (year)	1.07	1.04–1.10	<0.0001
Smoking during pregnancy	1.37	0.76–2.46	0.2902
Parity	0.65	0.56–0.76	<0.0001
Ethnicity			
Europe	REF		
Africa	1.86	1.23–2.84	0.0037
North Africa	0.92	0.61–1.39	0.6941
Asia	0.62	0.15–2.62	0.5166
Caribbean	1.20	0.66–2.20	0.5471
Pakistan India Sri Lanka	1.31	0.77–2.24	0.3266
Other	0.36	0.11–1.18	0.0912
Employment at beginning of pregnancy	1.08	0.79–1.47	0.6332

DIP—late-diagnosed (i.e., ≥22 weeks of gestation) diabetes in pregnancy; eDIP—early-diagnosed (i.e., <22 weeks of gestation) diabetes in pregnancy; eGDM—early-diagnosed gestational diabetes mellitus; GDM—late-diagnosed gestational diabetes mellitus; HIP—hyperglycaemia in pregnancy.

**Table 3 jcm-10-03904-t003:** Determinants of large-for-gestational infant in multivariable analysis.

	Odds Ratio	95% Confidence Interval	*p*-Value
Glycaemic status			
No HIP	REF		
eGDM	0.96	0.70–1.31	0.7759
eDIP	0.95	0.27–3.40	0.9379
GDM	1.34	1.01–1.78	0.0435
DIP	0.98	0.29–3.34	0.9694
Pre-pregnancy body mass index (kg/m^2^)	1.09	1.07–1.11	<0.0001
Age (year)	1.00	0.98–1.02	0.7094
Smoking during pregnancy	0.44	0.25–0.78	0.0053
Parity	1.14	1.04–1.24	0.0032
Ethnicity			
Europe	REF		
Africa	0.56	0.39–0.79	0.0009
North Africa	1.34	1.03–1.75	0.0287
Asia	0.78	0.33–1.83	0.5616
Caribbean	0.58	0.35–0.98	0.0397
Pakistan India Sri Lanka	0.59	0.38–0.92	0.0205
Other	1.48	0.95–2.31	0.0799
Employment at beginning of pregnancy	0.98	0.78–1.22	0.8361

DIP—late-diagnosed (i.e., ≥22 weeks of gestation) diabetes in pregnancy; eDIP—early-diagnosed (i.e., <22 weeks of gestation) diabetes in pregnancy; eGDM—early-diagnosed gestational diabetes mellitus; GDM—late-diagnosed gestational diabetes mellitus; HIP—hyperglycaemia in pregnancy.

## Data Availability

Some or all datasets generated during and/or analysed during the current study are not publicly available but are available from the corresponding author upon reasonable request.

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
