# Peer review of "Prognosis Associated with Sub-Types of Hyperglycaemia in Pregnancy"

_jcm, 2021, doi:10.3390/jcm10173904_

Round 1

Reviewer 1 Report

Cosson and Colleagues performed a well-designed study on sub-types of hyperglycemia in pregnancy and their outcomes. The paper address a very important clinical topic particularly for its application in clinical practice. 

Minor points:

1) why the Authors decide to consider normal unknown blood pressure values earlier than 22 gestational weeks?

2) in the first section of the results the Authors state that the study population has an overall higher risk of HIP. Data reported in table S1 are univariate comparison (not risk estimates), for this reason Authors should say that tha the study population has a higher prevalence of HIP.

3) Why the Authors did not included family history of diabetes in the multivariate analyses they have performed?

4) In the discussion section of the manuscript a comparison with others studies reporting results on predictors of neonatal outcomes should be done (e.g. PMID: 30221320).

5) In the discussion section Authors state that this is the first study describing DIP diagnosed after 22 WG subsequent to normoglycemia screening in early pregnancy. In this regard, did the authors also consider the studies produced by their own research group?

Reviewer 2 Report

I read with interest the manuscript entitled “Prognosis associated with sub-types of hyperglycaemia in pregnancy”. In this study the authors  compare pregnancy outcomes in 4,665 women according to the following  hyperglycaemia in pregnancy sub-types:  normoglycaemia, gestational diabetes mellitus (GDM) , diabetes in pregnancy (DIP), early-diagnosed (i.e., <22 weeks of gestation) GDM, and  early-diagnosed DIP (eDIP). Gestational diabetes mellitus is a common condition observed in a large population of pregnant patient. Early identifying women with hyperglycaemia is importnant to minimize maternal and nenonatal complications.

This study is technically very well-performed and the findings are very interesting and informative. However, in my opinion some minor points should be explained or changed:

  1. There are no data on the exclusion criteria from the study. I mean mainly patients with chronic hypertension. As the authors analyze the incidence of hypertensive disorders in pregnancy, such clarification is very important.
  2. As the authors analyze the frequency of fetal malformations in the eDIP group compared to DIP group, it is probably also worth referring to the data on the more frequent occurrence of malformations in the eGDM group compared to the NoHIP and GDM groups.
  3. The data on the frequency of insulin therapy (Lines 175 and 176) differ from the data presented in Table 1. This must be checked and corrected.

Reviewer 3 Report

The article "Prognosis associated with sub-types of hyperglycaemia in pregnancy" is an interesting document on an important issue, which is the disturbance of carbohydrate metabolism in pregnancy. Easily defines the sub-types of hyperglycemia in pregnancy. The conducted analysis provides very interesting conclusions regarding the relationship of well-controlled glycaemia with complications of pregnancy such as hypertension, congenital malformations and LGA. 

My comments are:

1) inconsistent data in table 1 and the text - line 175-176

2) line 306 - there should be sub-types?

3) I don't understand the sentence lines 318-322 - please correct
